# Potassium Ion Channels in Malignant Central Nervous System Cancers

**DOI:** 10.3390/cancers14194767

**Published:** 2022-09-29

**Authors:** Yasmin Boyle, Terrance G. Johns, Emily V. Fletcher

**Affiliations:** 1Telethon Kids Institute, Perth Children’s Hospital, 15 Hospital Ave, Nedlands, Perth, WA 6009, Australia; 2School of Biomedicine, The University of Western Australia, 35 Stirling Hwy, Crawley, Perth, WA 6009, Australia

**Keywords:** potassium ion channels, high grade glioma, glioblastoma, medulloblastoma, cellular plasticity

## Abstract

**Simple Summary:**

Malignant central nervous system (CNS) cancers are notoriously difficult to treat, with just one-third of patients surviving five years post-diagnosis. Their location within the brain and brainstem presents several challenges to successful treatment, including surgical inaccessibility and designing effective therapies that pass the brain’s protective barrier. Furthermore, high-grade CNS cancer is also prone to recurrence, and the secondary tumours that arise are highly resistant to treatment. Potassium ion channels are a class of transmembrane proteins involved in maintaining the electrical microenvironment of cells. In cancers, these proteins are known to play a significant role in the development of cancer hallmarks like proliferation, invasion, and adaptive drug resistance. Our review explores the relationship between potassium ion channel function and the progression of malignant CNS cancers. Targeting these proteins presents a promising, novel treatment strategy, with several FDA-approved potassium ion channel-targeting drugs already in clinical use for various CNS disorders.

**Abstract:**

Malignant central nervous system (CNS) cancers are among the most difficult to treat, with low rates of survival and a high likelihood of recurrence. This is primarily due to their location within the CNS, hindering adequate drug delivery and tumour access via surgery. Furthermore, CNS cancer cells are highly plastic, an adaptive property that enables them to bypass targeted treatment strategies and develop drug resistance. Potassium ion channels have long been implicated in the progression of many cancers due to their integral role in several hallmarks of the disease. Here, we will explore this relationship further, with a focus on malignant CNS cancers, including high-grade glioma (HGG). HGG is the most lethal form of primary brain tumour in adults, with the majority of patient mortality attributed to drug-resistant secondary tumours. Hence, targeting proteins that are integral to cellular plasticity could reduce tumour recurrence, improving survival. This review summarises the role of potassium ion channels in malignant CNS cancers, specifically how they contribute to proliferation, invasion, metastasis, angiogenesis, and plasticity. We will also explore how specific modulation of these proteins may provide a novel way to overcome drug resistance and improve patient outcomes.

## 1. Introduction

Intercellular communication is fundamental to maintaining homeostasis in a biological system, and this exchange is facilitated by proteins known as ion channels. Ion channels comprise one or more subunits that arrange to form a specialised pore, allowing ion flux across cellular membranes. Over 300 unique ion channel subtypes exist in human cells [1], each classified based on their selectivity for specific metal ions (K^+^, Na^+^, Ca^2+^, Cl^−^, etc.) and gating mechanisms. Potassium ion channels describe a family that comprises more than 90 proteins and include calcium- and sodium-activated potassium channels (K_Ca_, K_Na_), inwardly rectifying potassium channels (K_ir_), two-pore domain potassium channels (K_2P_), and voltage-gated potassium channels (K_v_ or VGKCs) [2]. The selective flow of metal ions across cell membranes enables intercellular signalling by regulating the membrane voltage potential [3]. Cellular voltage can become more positive (depolarisation) or negative (hyperpolarisation) in response to environmental stimuli. For example, upon VGKC activation, a net positive charge (K^+^) flows out of the cell, hyperpolarizing the membrane. This K^+^ flow is a crucial aspect of several essential cellular functions, including growth and proliferation, generation of action potentials, and cell cycle regulation [4].

One way that ion channels contribute to maintaining homeostasis is through regulating the resting membrane potential (RMP) of cells. The RMP of a cell refers to the difference in electrical potential across the plasma membrane when the cell is in a non-excited state (described in [5]). This is maintained primarily by the Na^+^/K^+^ ATPase pump, which pumps three Na^+^ ions out of the cell and two K^+^ ions into the cell to generate a concentration gradient across the plasma membrane. The RMP varies widely based on cell type and localisation; for example, mature astrocytes rest at approximately −80 mV [6], myocardial cells at −90 mV [7], and non-excitable red blood cells at −10 mV [8], while cancer cells, such as glioblastoma (GBM), typically exhibit much more depolarised RMP’s (−20 to −40 mV) [9]. A dynamic membrane potential allows excitable cell types (neurons and muscle cells) to experience rapid and significant changes in polarity, generating action potentials. However, the importance of ion channel proteins extends beyond action potential propagation, as they also play a significant role in non-excitable cell physiology, particularly regulating the cell cycle. The cell cycle describes a process where after cell division, the cell enters a growth phase (‘G_1_’), then undergoes DNA duplication (‘S’), followed by a second growth phase where the DNA is checked for errors (‘G_2_’) before mitosis (M). Mediated by ion channels, cyclic changes in voltage potential of the cell membrane accompany physical cellular changes during each phase (Reviewed in [10]). For example, there is a direct relationship between cellular electrical activity and mitosis, with Cone et al. [11] being the first to demonstrate that DNA synthesis can be induced through sustained cell depolarisation alone. Moreover, potassium ion channels are integral in progressing the cell cycle through various transition points, such as G1/S and S/G2. VGKCs, K_ir_, and K_Ca_ channels facilitate these transitions by altering membrane potential, regulating cell volume, and increasing Ca^2+^ influx [12,13]. Indeed, ion channels known to be involved in the cell cycle are upregulated in many cancer types (reviewed in [14]), increasing their proliferative potential and causing the dysregulation of cell growth, hallmarks of cancer pathology. 

Aberrant ion channel expression and dysfunction can give rise to various disease states, known as channelopathies [15]. Further, the close association between ion channels and cancer has led to the development of a new classification: oncochannelopathies (reviewed in [16]). Although not all cancer hallmarks are inherently linked to ion channel mutations, dysfunction, or abnormal expression, these can certainly contribute to disease progression. These hallmarks include unlimited proliferation, increased invasion and migration, and evasion of apoptosis [17]. Further, potassium ion channels (primarily K_Ca_) promote recruitment of existing vasculature (angiogenesis) and the formation of entirely new vasculature (vasculogenesis) [18,19,20]. This is essential in providing oxygenated blood carrying essential nutrients to both healthy and malignant cells. 

Previous authors have more generally reviewed the link between ion channel function and cancer progression [16,21,22,23] Therefore, we will focus specifically on how potassium ion channels contribute to central nervous system (CNS) cancers and explore their role in neoplastic cell transformation and growth. Evidence shows that malignant glioma cells essentially ‘hijack’ endogenous potassium ion channels to aid growth and metastasis. By upregulating the activity of these proteins, malignant cells can increase proliferation and alter their morphology to facilitate invasion [24,25]. In contrast, blockade of specific potassium ion channel subtypes results in apoptosis and growth arrest in numerous cancer cell lines, including GBM cells [26,27,28,29]. Despite the significant increase in the number of studies surrounding ion channel function in GBM in recent years, there remains significant gaps to bridge before we see ion channel targeted therapeutics in HGG treatment.

## 2. Potassium Ion Channels

The potassium ion channel family is the largest of all ion channel classes and comprises four subfamilies: voltage-gated (VGKC), sodium and calcium-activated (K_Na_, K_Ca_), inward-rectifying (K_ir_), and two-pore domain (K_2P_) potassium channels [30]. The channels in each subfamily differ based on their domain structure, gating mechanisms and function. Figure 1 provides a schematic representation of the general structure of these protein subfamilies. VGKC, K_Na_ and K_Ca_ channels share a basic 4 α-subunit structure with a single pore-forming region (TM5–TM6), while K_ir_ channels consist of only two α-subunits. K_2P_ channels possess two separate pore-forming domains and a four α-subunit structure. Furthermore, these multimeric proteins are often associated with one or more auxiliary β-subunits (encoded by genes *KCNE1*–*5*) that further influence channel gating [31].

VGKCs conduct K^+^ ions at a fast rate due to their highly conserved pore domain, which is closely linked to their function [32]. The pore consists of a channel gate and a selectivity filter, preventing other metal ions from entering the channel. K^+^ ions move in a single file arrangement through the channel pore, where the cumulative repulsion of charge-charge interactions between the positively charged K^+^ ions largely drives their movement [33]. VGKCs remain closed until they sense a depolarisation at the cell membrane, at which point the protein undergoes structural changes, resulting in an open conformation [34,35]. This allows K^+^ ions to flow through the pore and into the cell, until the membrane is repolarised, and the channel becomes inactivated. Because of their functional properties, VGKCs are essential to numerous biological processes that depend on sensing changes to membrane potential. This includes potential propagation [36], cellular growth and changes to volume [10], apoptosis regulation [37], and cardiac muscle function [38]. 

While K_Ca_ and K_Na_ channels may share structural parallels with their voltage-gated counterparts, they are functionally different. K_Ca_ channels are not voltage-dependant, and instead are gated by intracellular Ca^2+^. K_Ca_ channels are divided into three subfamilies: small conductance (SK), intermediate conductance (IK) and big conductance (BK) channels [42]. SK and IK channels are activated when calmodulin binds to their receptor domain in response to low intracellular concentrations of Ca^2+^ (~0.5 μM). BK channels differ in that they contain an additional transmembrane domain, allowing for the presence of two high-affinity Ca^2+^ binding sites [43,44]. Thus, BK channels are directly activated and opened by Ca^2+^ binding, bypassing the need for calmodulin. BK channels are expressed in many different tissues, and they serve several biological needs, including action potential duration and neurotransmitter release from presynaptic terminals (reviewed in [45]), as well as signalling between neurons and cerebral vasculature, controlling local cerebral blood flow [46]. This is critical, as normal brain functioning, especially during periods of increased neuronal activity, is dependent on sufficient local blood flow. It has been shown that astrocytic BK channels are activated following an increase in internal Ca^2+^ in states of heightened neuronal activity [47]. Upon activation, BK channels pump large quantities of K^+^ ions out of the cell, triggering K_ir_ channel activation in smooth muscle cells, resulting in hyperpolarisation. This causes vasodilation, or the widening of local vasculature, enhancing cerebral blood flow. 

The K_Na_ channel sub-family is represented by just two proteins known as Slack and Slick, encoded by *KCNT1* and *KCNT2*, respectively. These channels are activated by Na^+^ entry and there is a direct relation between their activity and intracellular Na^+^ concentration [48,49]. They are expressed throughout the CNS where they modulate the firing pattern and excitability of neurons [50]. Their structure mirrors that of K_Ca_ channels, but with an additional residues within the C-terminal region that regulate K^+^ conductance [40]. Due to their integral role in normal neuron function, *Slack* mutations that produce just a single amino acid change can result in epilepsy and intellectual disabilities [51]. However, the potential involvement of K_Na_ channels in cancer is yet to be explored. 

K_ir_ channels are primarily responsible for K^+^ influx and have a range of physiological roles depending on their tissue distribution. These channels remain open at rest and therefore play a key role in maintaining the RMP by regulating membrane voltage to stay close to the K^+^ Nernst potential (−60 mV to −90 mV). Additionally, K_ir_2.1 has been shown to play important roles in muscle contraction and early bone development [52,53]. Perhaps most importantly, K_ir_ channels are integral proteins in maintaining normal cardiac physiology, where they are responsible for slowing the heart rate [54]. This subset of K_ir_ channels, termed K_ACh_, are opened in response to binding with G-protein subunits. Stimulation of the vagus nerve triggers acetylcholine release, which in turn releases G-protein subunits. These bind to K_ACh_ channels, activating them and hyperpolarising the cell membrane, slowing action potential frequency and heart rate [55,56]. 

Finally, K_2P_ channels comprise an essential group of proteins in supporting neuronal function. These channels are unique in that they comprise two distinct pore domains within a singular protein. They are often referred to as ‘leak’ channels, as they produce relatively small, non-inactivating K^+^ currents across a wide range of membrane potentials [57]. K_2P_ channel currents can be influenced by a number of stimuli, including kinases, lipids, G-proteins, changes in pH, mechanical force, and interactions with other proteins [58]. However, they are markedly more resistant to traditional potassium ion channel antagonists such as tetraethyl ammonium, 4-aminopyridine and several biological toxins. This increased resistance could be attributed to the presence of a unique protein loop ‘cap’, that may shield the pore domain from antagonist binding [59,60]. Some have argued that this feature has caused K_2P_ channels to be largely overlooked as drug targets, despite their expression in many tissues and role in chronic pain, migraines, and cancer (reviewed in [61]).

### Potassium Ion Channels in Non-CNS Cancer

Numerous studies have identified dysregulated K^+^ channel expression across various cancer types, summarised in Table 1 (reviewed in [23]). From these data, it is clear that potassium ion channels are dysregulated across many different malignancies, and that this may have downstream effects on cancer hallmarks. Interestingly, higher K_v_10.1 (also known as human ether-a-go-go 1 or EAG1) expression is found in over two-thirds of cancer types throughout various tissues [62]. In healthy tissue, it is expressed throughout various regions of the brain, including the hippocampus, hypothalamus, and striatum [63]. K_v_10.1 is also transiently expressed during myoblast differentiation [64]. It was then established that K_v_10.1 likely plays a role in cellular proliferation [65,66], which was confirmed by Pardo et al. in 1999 [67]. In vitro, non-malignant Chinese hamster ovary (CHO) cells transfected with K_v_10.1 showed significantly increased growth over wild-type cells, as well as increased metabolic activity. When implanted into mice, the CHO/K_v_10.1 cells induced the growth of highly aggressive, necrotic tumours. They also identified endogenous expression of K_v_10.1 in human breast, cervical and carcinoma cancer cell lines, but not in equivalent healthy tissues. This was confirmed by Hemmerlein et al. [62], who observed extensive K_v_10.1 expression in human breast, lung, prostate, and liver tumours, despite the fact that K_v_10.1 is typically very limited outside the CNS in healthy tissues. These findings were instrumental in establishing the role of potassium ion channels, particularly VGKCs, in cancer.

There are several proposed ways that VGKCs influence cell proliferation in both healthy and malignant cells (reviewed in [25,41,105]), as shown in Figure 2. First, VGKCs play a central role in the oscillations in membrane polarisation that occur throughout the cell cycle, such as the membrane hyperpolarisation that occurs at G_1_ (top panel, Figure 2). When an efflux of K^+^ ions occurs, the cell membrane is hyperpolarised, causing an influx of Ca^2+^ ions. This, in turn, triggers signalling pathways that release further Ca^2+^ from internal stores, increasing the intracellular Ca^2+^ concentration and promoting proliferation via calcium signalling. 

Evidence suggests that VGKCs also play an integral role in migration and invasion (middle panel, Figure 2). VGKCs, particularly K_v_10.2 channels, regulate cellular volume and shape via the influx/efflux of K^+^ ions [106]. As these ions are pumped out of the cell, internal solute concentration decreases, allowing the osmotic movement of water out of the cell and reducing cell volume. This process occurs in different parts of the cell membrane, resulting in localised changes to cellular shape and volume, thus allowing malignant cells to vary their size to navigate tight extracellular spaces [106,107]. 

Finally, potassium ion channels have been implicated in the pro-oncogenic process of angiogenesis due to their crucial role in cell signalling and localisation at the cell membrane (bottom panel, Figure 2). K_v_10.1 has been directly linked to pro-angiogenic pathways in CNS and blood cancers [108,109]. In these tumours, K_v_10.1 is directly responsible for the secretion of vascular endothelial growth factor (VEGF), a pro-angiogenic protein responsible for the formation of blood vessels. Further, a subset of K_ir_ channels known as K_ATP_ channels (activated by increased ATP) have been established as contributors to angiogenesis [110]. Direct pharmacologic activation of these channels induced angiogenic effects, including endothelial cell migration and network formation, both in vitro and in vivo. 

## 3. Malignant CNS Cancer

Cancer has long been a leading cause of mortality worldwide and resulted in an estimated 10 million deaths worldwide in 2020 alone [111]. This review will focus on cancers of the CNS, primarily gliomas. These cancers are graded by the World Health Organisation (WHO) based on their degree of malignancy. Grade I and II gliomas are classified as ‘low-grade’, with generally favourable prognostic outcomes and less likelihood of recurrence following debulking surgery [112]. Pilocytic astrocytoma, diffuse astrocytoma, and ganglioglioma are some examples of the many subclassifications of low-grade glioma (LGG). Grade III and IV gliomas are considered high-grade, carrying significantly worse patient outcomes and a high likelihood of tumour recurrence. 

In 2020, the worldwide incidence rate for primary malignant CNS tumour diagnosis was 3.5 per 100,000 [111]. Gliomas make up many of these cases, accounting for approximately 78.3% of malignant CNS cancer diagnoses in the United States [113]. With hundreds of thousands of new cases diagnosed each year, there is a significant burden placed on the healthcare system, and individual patients, to treat this disease. For example, the median expenditure for an individual undergoing treatment for HGG is $184,159.83, with radiation therapy accounting for most of this cost [114]. Only one-third of malignant CNS cancer patients will reach the 5-year survival point post-diagnosis [113]. This high mortality rate is largely due to inadequate treatment protocols.

GBM, diffuse midline glioma (DMG), and anaplastic astrocytoma are classified as HGG. GBM carries the worst prognosis of all adult primary CNS cancers, with a 5-year survival rate of less than 5% and an average survival time of just 15 months [115]. GBM accounts for almost 50% of all malignant primary CNS tumours [113] and the average age of GBM patients at diagnosis is 64 years. The cell type of origin for GBM has remained a debate for decades, with a current consensus that neural stem cells (NSCs), NSC-derived astrocytes and oligodendrocyte precursor cells (OPCs) are all cells of origin [116]. Current treatment strategies have remained unchanged for many years and are considered primarily palliative, beginning with surgical resection of as much of the tumour mass as is feasible (Figure 3). Due to GBM’s highly infiltrative and diffuse growth pattern, it invades surrounding healthy brain parenchyma to a high degree, making whole tumour debulking impossible [117]. Surgery is then followed by combined adjuvant radiotherapy and temozolomide treatment, which has shown to increase the 2-year survival rate from 10.4% to 26.5% [115]. Even with these combined efforts, the prognosis for patients with GBM remains extremely poor, with treatment often adding mere months to their survival time.

LGG describes a diverse group of grade I and II primary brain tumours that account for approximately 17.9% of newly diagnosed malignant CNS cancer cases in the United States [113]. These patients have a median age of 41 years and a median survival of approximately seven years following diagnosis and intervention [118]. Despite having a longer overall survival time than HGG, LGG is still considered a generally fatal disease because all grade II gliomas will eventually progress to grade III or IV. Grade I LGG is sometimes grouped separately to grades II, III, and IV as these typically present as benign tumours in children and are often cured following surgical resection [119]. Treatment for grade II LGG is generally addressed in four stages: observation, surgery, radiation, and chemotherapy [120].

Finally, medulloblastoma (MB) is the most common form of brain cancer in children, accounting for nearly 10% of all childhood CNS tumours [121]. Approximately 500 new cases of MB are diagnosed each year in the United States, and survival times are heavily dependent on the molecular subgroup with which a patient is diagnosed. The non-Wnt/SHH tumour subgroup 3 has the worst prognosis, accounting for approximately 30% of cases, with median paediatric 10-year survival rates of only 50%. MB treatment follows a similar path to glioma, consisting of surgery, radiation therapy, and adjuvant chemotherapy. These procedures can cause devastating, lifelong side effects to paediatric patients, and trauma associated with undergoing treatment. Successful implementation of targeted therapy could bypass the need for harmful radiation and chemotherapy for these patients, greatly increasing their quality of life and survival.

### 3.1. Targeted Therapies

As discussed, current treatment methods for HGG are non-curative, hence the pressing need to develop novel and improved treatment strategies. Several challenges exist in treating HGG through traditional methods (reviewed in [122]). Firstly, their location within the brain or brainstem reduces accessibility for complete tumour resection and increases the likelihood of side effects from tumour growth into healthy brain tissue. In addition, HGG tumours are highly heterogenous, and often constitute multiple different sub-populations of cells within a single tumour. Furthermore, many existing chemotherapeutics do not cross the blood-brain barrier (BBB) and present many undesirable side effects to patients due to their non-targeted mechanisms of action. Targeted therapy represents an attractive solution to these problems, through which malignant cells are specifically targeted, bypassing healthy brain tissue and reducing off-target effects. However, many targeted therapies have tried and failed to improve survival in patients with HGG, despite promising results in vitro and in vivo (reviewed in [123]). 

Epidermal growth factor receptor (EGFR) and its variant EGFRvIII are commonly amplified in GBM, with over half of primary GBM cases showing upregulated EGFR [124]. This has led to a number of clinical trials directed at EGFR inhibition to prevent or slow GBM tumour growth. Rindopepimut was developed as a novel ‘tumour vaccine’ to target the EGFRvIII mutation site and was put through phase I and II clinical trials [125]. Primary GBM patients whose tumours expressed EGFRvIII were recruited and treated with a combination of rindopepimut and temozolomide as standard. Patient outcomes varied across the group but were generally more favourable than matched historical controls. Trial participants overall survival ranged from 22–26 months compared to 15 months with TMZ treatment alone. This represented a promising breakthrough in targeted therapy for HGG. Unfortunately, the rindopepimut and temozolomide combination therapy failed to improve survival in a phase III trial [126].

Factors influencing angiogenesis are also primary targets for HGG treatment, as reducing blood flow to malignant cells can arrest growth and invasion. Yet, the anti-angiogenic agent cediranib, a VEGF receptor tyrosine kinase inhibitor, was ineffective in influencing GBM patient outcomes in phase III trials [127]. While bevacizumab, a monoclonal antibody that binds to VEGF-a, also failed to improve overall survival in several phase III trials [128,129]. There are several potential reasons underpinning the failure of targeted therapies in HGG treatment. These include inadequate drug penetration to the tumour site, tumour heterogeneity, lack of tumour dependence on target proteins, clonal evolution, and antigen escape [123]. Further, because HGG cells possess a high degree of plasticity, reversibly shifting between multiple unique ‘states’ [130] HGG cells bypass targeted therapies by adapting alternative growth pathways and maintaining proliferation. As ion channels are primary drivers of cellular plasticity [131,132], inhibiting these proteins could reduce resistance to targeted therapies.

Regarding LGG, a recent study identified a potentially novel and effective therapy for paediatric BRAF V600 mutation positive LGGs [133]. These BRAF-mutated tumours are particularly resistant to the current standard of care, consisting of treatment with carboplatin and vincristine. This study saw over 50 patients treated with dabrafenib and trametinib, compared against a control group receiving standard LGG chemotherapy. 47% of patients in the experimental group responded significantly to treatment, showing markedly reduced tumour size, if not complete remission. Recurrence was prevented for an average of 20 months in the experimental group against just seven months in the control. Similarly effective targeted therapies against LGG could greatly benefit patients in preventing the progression of the disease into a more malignant phenotype. 

The effectiveness of targeted therapies against MB has been evaluated both in vitro and in vivo, with varying levels of success. One example is vismodegib, an FDA-approved inhibitor of the tumour-driver *SMO*, an integral gene in the progression of the most malignant forms of MB. Unfortunately, this gene is commonly mutated, preventing adequate drug binding [134]. As a further complication, the drug was found to cause severe growth plate fusions in younger patients due to the involvement of *SMO* in development [135]. Finally, a combination of vismodegib and temozolomide failed in a stage II clinical trial when no significant benefits were observed in the experimental group [136]. Another attractive target for MB treatment is the intracellular lipid kinase PI3K, as it plays a key role in the MB cell signalling and downstream activation of other kinases, including mTOR. This regulates cancer hallmarks such as growth and cell survival [137]. A recent in vivo study showed that treating childhood MB cell lines with PI3K inhibitors alone and in combination with cytostatic drugs significantly reduced proliferation [138,139]. Several clinical trials are currently underway to evaluate the safety and effectiveness of PI3K modulators and other targeted therapeutics in MB (summarised in [140]).

### 3.2. Cellular Plasticity and Drug Resistance

Both healthy and malignant neural cells are highly plastic, a significant barrier preventing the success of targeted HGG therapy. Sturm et al. [141] identified the six distinct states, differentiated by their DNA methylation patterns, between which HGG cells continually and reversibly transition. This plasticity underpins adaptive drug resistance, in which alternative signalling pathways are activated in response to the targeted blockade of integral proteins [142,143,144]. This process largely relies on glioma stem cells (GSCs), unique malignant cell populations capable of forming heterogenous HGG tumours. Many cancerous cells that remain following standard HGG treatment can be classified as GSCs. They are characterised by their heightened resistance to drugs, unlimited proliferation, and multipotent differentiation [145]. These cell populations have been identified across all grades of glioma, with Bourkoula et al. [146] proposing that GSC features have a substantial prognostic value in LGG, specifically. Yet, most studies investigating GSCs focus on HGG. This is likely because GSC populations are more highly expressed in HGG, and isolation techniques and culture conditions are generally optimised for GSCs derived from HGG rather than LGG [147].

GSCs are believed to be responsible for HGG’s adaptive response to treatment by developing alternative oncogenic pathways that bypass the targeted proteins or pathways [148,149]. For example, TMZ resistance is common in GBM tumours, and GSCs are believed to be primarily responsible for this (reviewed in [150]). Firstly, whole genome sequencing of matched primary and recurrent GBM tumours showed common deleterious mutations in *TP53* that originated from GSC populations [151,152]. Mutations in *TP53* are common in recurrent GBM, and linked to heightened invasion, migration, proliferation and GSC propagation [153]. Orzan et al. [152] identified that these distinctive *TP53* mutations in recurrent tumours were detectable at a lower rate in the matched primary tumours, indicating that these GSC populations already existed, but were expanded under therapeutic pressure induced by TMZ. Following TMZ treatment, the most resilient GSCs remain and propagate recurrent, heterogeneous GBM tumours with heightened drug resistance. Despite the DNA synthesis blockade induced by TMZ, GSCs stimulate the release of growth factors and cytokines to further tumour growth. A similar phenomenon is observed with targeted treatments, one such example being dendritic cell vaccination [154]. Following vaccination of six GBM patients, tumours shifted their cellular states towards a more proneural and proliferative state (compared to patients receiving TMZ). These states upregulate genes controlling migration and invasion, as well as DNA replication and cell cycle regulation, bypassing the anti-proliferative effect of dendritic cell vaccination and furthering oncogenesis. 

It has been established that ion channels, including potassium ion channels, are key proteins in stem cells, including GSCs. Following treatment with the K_Ca_3.1 antagonist TRAM-34, GSC cell mobility and migration was significantly reduced [155]. Similarly, BK channels showed greater expression in GSC cells vs. comparable control glioma cells and inhibition of this channel reduced migration of GSC cells in vitro [156]. Further, Pchelintseva et al. [157] detailed a model wherein K_Ca_ channels are fundamentally involved in stem cell differentiation. At the relatively depolarised RMP of stem cells (−20 to −30 mV), intercellular Ca^2+^ is increased. This, in turn, activates K_Ca_ channels, leading to K^+^ efflux and membrane repolarisation. In response, Ca^2+^ efflux increases, suppressing K_Ca_ activity once again. This cycle continues, causing oscillations in Ca^2+^ concentration within stem cells. Depending on the amplitude and frequency of these fluctuations in Ca^2+^, stem cells can trigger a range of cellular behaviours involved in differentiation. Fundamentally, this includes initiation of gene expression leading to a shift in cell type, to phenotypic effects like increased migration. K_Ca_ channels (including BK) play a unique and integral role in several cancer hallmarks. Hence, developing treatments that target these key proteins in GSCs could present a unique opportunity to hinder stem cell plasticity and reduce evasion of targeted therapy. 

Similarly, to glioma, MB tumours are largely heterogenous and diverse, exhibiting various cell types, including stem cell-like populations (MBSCs) [158]. Single-cell analysis of MB tumours following vismodegib treatment revealed an increase in the population of MBSCs, which express the oncogene, *SOX2* [159]. *SOX2* plays a significant role in cancer cell stemness and is often overexpressed in the stem-cell populations of both HGG and MB [160]. There is a positive correlation between malignancy grade and *SOX2* expression in brain cancer, making it an attractive target for novel therapeutic strategies. It has also been established as a driver for MB tumour recurrence, given that MBSCs expressing *SOX2* are enriched following treatment [159].

Interestingly, Metz et al. [161] have identified that *SOX2* expression is specifically optimised for tumour growth and that either knockdown or elevation of its’ expression inhibits MBSC proliferation. Elevating *SOX2* expression halted MB tumour growth and returning expression to endogenous levels allowed growth to resume in vitro. Despite their resistance to *SMO* inhibitors like vismodegib, recent studies have shown that MBSCs may be sensitive to treatment with GLI (glioma-associated oncogene) modulators [162]. These proteins are downstream targets of *SMO*, and typically function as transcriptional activators. Pharmacological targeting of GLI-depleted MBSC sub-populations from MB tumours, both in vivo and in vitro. This is a promising result, as MBSCs represent a significant barrier to successful MB treatment. Further investigation into the importance of the GLI pathway may represent a novel treatment strategy for recurrent MB. 

## 4. Potassium Ion Channels in Malignant CNS Cancer 

As discussed, potassium ion channels play an essential role in cancer hallmarks like proliferation, invasion, and angiogenesis, which are key drivers of tumour progression in CNS cancers. Here, we will examine how different potassium ion channel subtypes contribute to oncogenesis in HGG, LGG, and MB.

### 4.1. High-Grade Glioma

The K^+^-regulating ion channel family is perhaps the most well understood regarding HGG pathophysiology and tumourigenesis, although several published studies have conflicting results. For example, BK channel overexpression has been identified in biopsy samples of GBM patient tumours, with the level of overexpression directly related to increasing malignancy grade [163]. The importance of BK channels was further solidified by the discovery of a novel splice isoform of the protein (gBK channels) that is exclusively expressed in GBM. The gBK channels appear to be directly involved in the cellular volume/size changes required for invasion of healthy brain parenchyma. Increased cytosolic Ca^2+^ concentration, for example via treatment with menthol, activates gBK channels and facilitates greater GBM cell migration and invasion [164]. While siRNA-induced knockdown of gBK channels in GBM does not change proliferation rates [165], various BK channel inhibitors can decrease cellular proliferation and increase tumour shrinkage in GBM cell cultures [166]. Conversely, BK channel activators were shown to reduce glioma cell migration by as much as 50%, suggesting that increased BK channel activity may, in fact, reduce glioma invasiveness [167]. 

Inwardly rectifying K+ (K_ir_) channels have also been implicated in GBM progression and tumourigenesis. K_ir_ channels are found in abundance in normal glial cells, where they produce large, inwardly rectifying K^+^ currents that stabilise the membrane potential to approximately −90 mV [168]. However, when glial cells become malignant, the cellular membrane becomes depolarised, increasing up to −20 mV [9]. Further studies showed that K_ir_ channels are expressed in GBM but are localised to the nucleus rather than the plasma membrane [169]. This reduces the effect that K_ir_ channel function can have on overall cellular membrane potential. It may also contribute to epileptic seizures in many patients with GBM, as mislocalisation of K_ir_ proteins can result in increased extracellular K+ ion concentration, which is associated with spontaneous seizures [170]. Thus, although K_ir_ channels may not be as directly related to tumourigenesis as some other ion channel subtypes, they may represent potential targets in treating GBM-induced seizures. 

A recent study [171] hypothesised that a novel mechanism involving both BK and K_ir_4.1 channels might be partly responsible for the highly invasive nature of GBM cells. K_ir_4.1 channels are normally highly expressed in healthy glial cells but are noticeably downregulated in GBM tumours [169]. The authors proposed that a small fraction of highly active K_ir_4.1 channels remain continuously open, causing K^+^ ion efflux and a more depolarised RMP (~30 mV) leading to the transient activation of BK channels and significantly increasing cytosolic water efflux. Together, these processes allow GBM cells to rapidly reduce their internal solute and water concentration, facilitating the changes in cell shape and size required to navigate tight extracellular spaces during migration.

The VGKC class is the largest of the main four potassium ion channel classes; however, only a small proportion of VGKC subtypes have been implicated in promoting GBM growth. The first is K_v_1.3, which shows decreased plasma membrane expression but increased mitochondrial expression in GBM cell lines. Pharmacological inhibition of K_v_1.3 channel function results in significantly increased apoptosis across the cell population. Unfortunately, these findings did not translate to in vivo tumour models, wherein treatment with the same VGKC-blocking drugs did not induce apoptosis in mice [29]. Specific K_v_1.3 inhibition using the drug 5-(4-phenoxybutoxy)psoralen (PAP-1), can reduce GBM cell invasion, possibly by preventing critical changes to cellular volume [172]. These findings were successfully translated to in vivo mouse models, in which PAP-1 treatment resulted in an up to 70% decrease in tumour size, compared with untreated mice. Thus, it will be of great interest to see how these discoveries relating to K_v_1.3 function in GBM progress over time. 

Expression of another member of the VGKC family, K_v_1.5, shows an inverse relationship to increasing malignancy grade in glioma. High expression is observed in lower-grade gliomas such as astrocytoma, with the lowest expression detected in GBM tumour samples [173]. K_v_10.1 appears to have a more integral role in GBM development. This channel was positively expressed in 85.3% of tumours in a cohort of 75 GBM patients. Furthermore, GBM patients with brain metastases, but low levels of K_v_10.1 expression, had significantly longer survival times than those with high K_v_10.1 expression [174]. Further, a 2016 study [175] identified that K_v_10.1 suppression sensitised GBM cells to treatment with TMZ in vitro. Pharmacological inhibition of K_v_10.1 (using astemizole) with TMZ decreased GBM cell viability by 77%, compared to just 46% with TMZ treatment alone. Similarly to Kv10.1, K_v_11.1 channels are frequently highly expressed in cancer cells (e.g., [176]). K_v_11.1 is highly expressed in HGG, but not LGG, and these channels promote the secretion of the pro-angiogenic protein VEGF [108]. Some evidence suggests that pharmacological blockade of K_v_11.1 can induce apoptosis in GBM cell lines in a concentration-dependent manner [177]. This could be due to the integral role that K_v_11.1 channels play in malignant cell cycle regulation, where they may be responsible for limiting the depolarised RMP of cancer cells. This allows for larger hyperpolarisations, driving the Ca^2+^ ion flux required for cell division [178]. 

Potassium ion channel expression may also be of prognostic value in GBM. Wang et al. [179] used RNA sequencing data derived from paediatric GBM tumours to identify genes associated with better or worse patient outcomes. These data were compiled from three databases, The Cancer Genome Atlas (TCGA), The Compendium of Cancer Genome Aberrations (CCGA), and The Repository of Molecular Brain Neoplasia Data (REMBRANDT). They identified three potassium ion channel genes associated with patient prognosis in paediatric GBM: *KCNN4*; *KCNB1*; and *KCNJ10* (encoding K_Ca_2.4, K_v_2.1, and K_ir_4.1, respectively). This three-gene signature significantly correlated with malignant progression and overall patient survival across all databases. However, the scope of this study was limited based on the relatively small population of paediatric samples in the patient datasets. 

Finally, Venkatesh et al. [180,181] recently identified an important role for K^+^ currents in a rare and deadly form of paediatric HGG-diffuse midline glioma (DMG). They successfully showed that neuronal activity directly influenced glioma growth via K^+^ ion signalling. Rapid and sustained action potential generation from peritumour neurons resulted in increased K^+^ within the extracellular space. This increased depolarizing K^+^ currents in the glioma cells, driving cell proliferation and growth. Treatment with barium, a blocker of inwardly rectifying K^+^ currents, entirely diminished these currents, identifying that K_ir_ channels are responsible for this intercellular signalling. 

Research surrounding the potential integral role of VGKCs in HGG growth and metastasis is still in its infancy. However, given the importance of VGKC functions in other cancer types (Table 1), and promising data with regard to glioma, there is potential for VGKCs to act as novel therapeutic targets for HGG.

### 4.2. Low-Grade Glioma

Compared to HGG, there is significantly less information available regarding how potassium ion channels contribute to LGG disease progression. However, a few studies have explored how specific potassium ion channel subtypes are expressed in LGG. Diffuse grade II *IDH1*-mutant gliomas are particularly rare, and often affect young adults. These tumours are particularly heterogenous, displaying multiple different cell subtypes within a single tumour, but the underlying mechanisms of this were not well understood. Augustus et al. [182] identified that multiple proteins were involved in developing this intratumoural diversity, including K_Ca_2.3. In the astrocyte-like tumoural cell populations, electrically active K_Ca_2.3 channels were highly expressed, while the oligodendrocyte-like tumour cells showed significantly lower expression. This suggests that K_Ca_2.3 may have potential to be used as a marker for astrocyte-like tumoural cells, aiding in patient tumour classification and therapeutic strategies.

In a study involving nine unique diffuse astrocytoma samples (grade II) and 51 HGG samples (grade III and IV), K_v_1.5 expression was found to be significantly higher in cases of LGG (22.2% vs. 10.7%) [183]. Preussat et al. [173] also identified an inverse relationship between malignancy grade and K_v_1.5 expression, which was highest in astrocytomas, moderate in oligodendromas, and lowest in GBM. A similar pattern was observed in lymphoma cases, where K_v_1.5 was expressed at a decreased rate in highly malignant samples [184]. Similarly, K_v_10.1 and 11.1 mRNAs are more highly expressed in LGG (pilocytic and diffuse astrocytoma) and healthy brain tissue than in HGG [178]. 

Our understanding of why this may be the case is limited; however, it could be due to the upregulation of other potassium ion channels (such as gBK and K_v_ subtypes) in more malignant cancers. gBK channels possess a large conductance, efficiently transporting K^+^ ions, potentially reducing the need for the relatively smaller K^+^ conductance of K_v_1.5 channels. As previously mentioned, gBK channels play an essential role in HGG tumourigenesis, contributing to proliferation and migration. Significant BK channel expression has also been recorded in low grade astrocytoma samples [185]. Further studies are needed to determine whether potassium ion channels are critically involved in LGG progression. Although LGG is generally associated with a more favourable patient outcome than HGG, grade II tumours will inevitably progress to grade III or IV. Thus, early intervention to treat LGG with novel ion channel therapies could prevent this progression to a more malignant phenotype.

### 4.3. Medulloblastoma

The majority of existing studies targeted at potassium ion channels and medulloblastoma have centred around the K_v_11.1 subtype. Huang et al. [106] were the first to observe that K_v_11.1 was overexpressed in MB tumours samples derived from various subgroups compared with healthy brain. K_v_11.1 knockdown reduced tumour cell growth in vitro and reduced tumour burden in xenograft mouse models. K_v_11.1 was confined inside the cell before the G2 phase of the cell cycle, and upon reaching late G2, it was trafficked to the plasma membrane. Therefore, K_v_11.1 knockdown induced G2 arrest, resulting in mitotic catastrophe and cell death. Further studies established that K_v_11.1 was specifically enriched in the most invasive subpopulations of MB cells, where it regulated changes to cell shape and volume, thereby facilitating migration [186]. This process is facilitated by the chloride ion channel, CLIC1, during which both proteins aggregate at lipid rafts to regulate cell volume [187]. Simultaneous knockdown K_v_11.1 and CLIC1 resulted in the suppressed growth/tumour burden in human MB cell lines and fruit fly models. Similarly, loss of the VGKC subtype K_v_2.2 results in significantly improved survival in mouse models of MB [188]. 

As previously described, MB tumours have localised subpopulations of drug resistant MBSCs, presenting a major roadblock to successfully treating the disease [189]. These cells are self-renewing and sustain tumour growth, leading to therapeutic resistance and eventual MB recurrence. K_v_2.2 is specifically enriched in MBSCs, and downregulation of this channel significantly reduces MBSC populations. These data suggest that K_v_2.2 may be of therapeutic value in treating MB as targeting critical proteins in MBScs could prevent growth, drug resistance, and eventual tumour relapse.

## 5. Potassium Ion Channels as Therapeutic Targets

Ion channels represent one of the most common targets of currently approved drugs, second only to G-coupled protein receptors [190]. Several potassium ion channel antagonists, categorised as class III antiarrhythmic agents, have been developed to treat cardiac arrhythmias. These include dofetilide, amiodarone, sotalol, and ibutilide, which act on the K_v_11.1, 11.2, and 11.3 VGKCs, respectively, but have known off-target actions other proteins [191]. It is well established that K_v_11 channels are responsible for mediating the heart’s inwardly rectifying K^+^ current. Furthermore, missense mutations in these channels can result in type 2 long QT syndrome, triggering arrhythmias [192]. Due to their integral role in maintaining K^+^ homeostasis in the heart, these drugs have potential serious adverse effects and can cause proarrhythmia [193]. Alternatively, the VGKC antagonist dalfampridine is used to aid multiple sclerosis patients with motor function and has a wide range of action on subtypes K_v_1.1 to 1.7, K_v_2.1 and 2.2, K_v_3.1 to 3.3 and K_v_4.1 and 4.2 [191]. Due to dalfampridine’s non-specificity for K_v_11 channels, it is not likely to cause these same adverse effects as class III antiarrhythmics. 

Not all ion channel drugs block channel function. A small group of FDA-approved drugs work by forcing the channel to remain open, allowing increased K^+^ ion flow. Most of these drugs, including minoxidil (targets K_ir_1.1) and diazoxide (targets K_ir_6.2), are classed as vasodilators and are used to regulate blood pressure, while ezogabine targets K_v_7.2 to 7.5 and acts as an anti-convulsant [191]. Alteration of VGKC function via blockade or sustained opening of the channel is of clinical value across several conditions. This is promising regarding CNS cancer treatment because these malignancies appear to be reliant on VGKC function for both growth and metastasis. Furthermore, because many potassium ion channel antagonists are already approved for clinical use in humans, there is potential for rapid clinical translation via drug repurposing. This includes the class III antiarrhythmics, as well as nateglinide and repaglinide, used in the management of diabetes [194]. When an existing drug is repurposed for a new function, such as cancer treatment, the process involved in receiving FDA approval for the new indication/s is vastly shorter than it would be for an entirely new molecule. With further research, there is potential for developing novel drugs that target specific potassium ion channel subtypes in several CNS cancers. 

## 6. Conclusions and Future Directions

Thanks to the collaborative efforts of numerous research teams, our understanding of the roles of ion channels in tumourigenesis has grown substantially in recent decades. The discovery that ion channels act as crucial drivers of homeostasis via proliferation and cell cycle regulation, and not just as propagators of action potentials, provided the basis from which much of our understanding has grown. It is now understood that ion channels are integral to the processes underlying cellular plasticity, the primary hurdle preventing the successful and curative treatment of HGG. Malignant HGG cells repurpose the endogenously expressed ion channels in healthy glial cells to further their development. Potassium ion channels are perhaps the most well-studied ion channel class in HGG oncogenesis, with many individual studies demonstrating how these cells utilise VGKC, K_Ca_, and K_ir_ channels for enhanced growth and metastasis. Several VGKC-targeting drugs have already received FDA approval, proving that alteration of their normal activity is possible without causing unwanted off-target effects. 

Further research should aim to elucidate the roles that potassium ion channels play in CNS cancers and to understand the cellular pathways and downstream proteins they influence. siRNA or CRISPR-mediated knockout studies involving key potassium ion channel subtypes in GBM cell lines would provide further insight into their involvement in key malignant cellular processes. Electrophysiological studies could also help determine whether or not these channels act independently of their current-modifying ion channel functions. Pharmacological inhibition of specific subtypes using repurposed drugs may also demonstrate the potential therapeutic value of these channels in GBM treatment. Finally, the development and use of accurate in vivo models of highly malignant tumours would be of great value, as studies involving cell lines alone can be somewhat limited in scope and potential.

Cancer represents one of the biggest threats to public health and has long been a pivotal contributor to mortality rates worldwide. CNS cancers represent a small percentage of newly diagnosed cancer cases annually yet make up a significant proportion of cancer-related deaths yearly. There is an urgent need to develop novel and improved treatment strategies for these patients, who currently have almost no hope of surviving an HGG diagnosis, regardless of their age or lifestyle. Numerous studies have implicated potassium ion channels as central to cancer progression; however, we are yet to see the development of an ion channel-based therapy for this disease. Thus, further exploration of the potassium ion channel subtypes preferentially expressed in CNS cancer could provide insight into the oncogenic process of this highly aggressive and invasive disease.

## Figures and Tables

**Figure 1 cancers-14-04767-f001:**
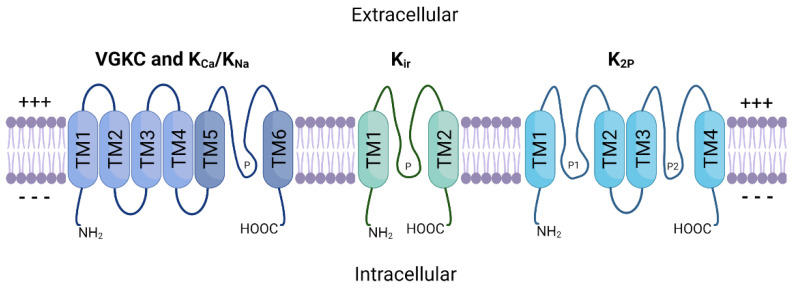
General schematic structure of the four major potassium channel subfamilies. TM1–TM6 represent transmembrane segments and ‘P’ represents pore-forming regions between subunits. VGKC and K_Ca_ channels share a common four subunit structure (TM1–TM4) that comprises the selective voltage-sensing domain [32]. TM5 and TM6 correspond to the pore domain found in all K^+^ channels [39]. K_Na_ channels also follow this basic transmembrane structure but encode additional residues within the C-terminal region that regulate K^+^ conductance [40]. K_ir_ channels have two transmembrane domains and a single pore-forming region between them. K_2P_ channels are comprised of four TM domains and two separate pores. Modified from [41].+++: Positively-charged side; ---: Negatively-charged side.

**Figure 2 cancers-14-04767-f002:**
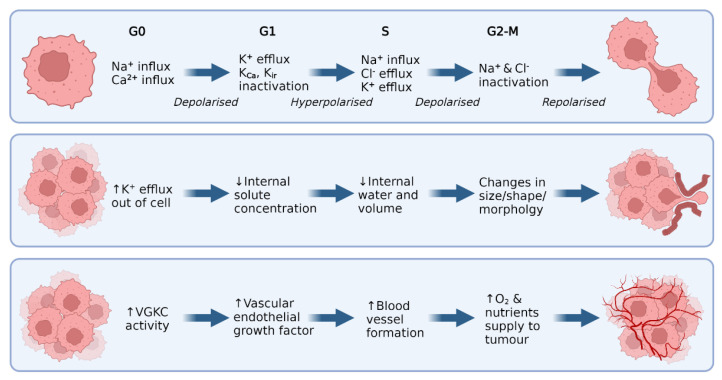
Potassium channels are involved in key cancer hallmarks in malignant cells. **Top panel:** To initiate the cell cycle (G0/G1) an influx of Na^+^ and Ca^2+^ occurs, depolarising the cell. In G1, VGKCs, K_ir_, and K_Ca_ channels activate and pump K^+^ ions out of the cell, hyperpolarising the cell. Towards the end of G1, all K^+^ channels are inactivated, and the cell is pushed into S phase. Here, Na^+^ is pumped into the cell, and Cl^-^ and K^+^ are pumped out. The now depolarised cell enters G2/M, where Na^+^ and Cl^-^ channels are inactivated, the cell is repolarised, and mitosis occurs. **Middle panel:** Malignant cells undergo K+ efflux to facilitate the water loss required to alter size and shape. This allows cancer cells to navigate tight extracellular spaces and invade tissue such as the brain parenchyma. **Bottom panel:** VGKCs, such as K_v_10.1, can induce VEGF secretion, thus stimulating the formation of vasculature to feed the growing tumour.↑: Indicates a relative increase; ↓: Indicates a relative decrease.

**Figure 3 cancers-14-04767-f003:**
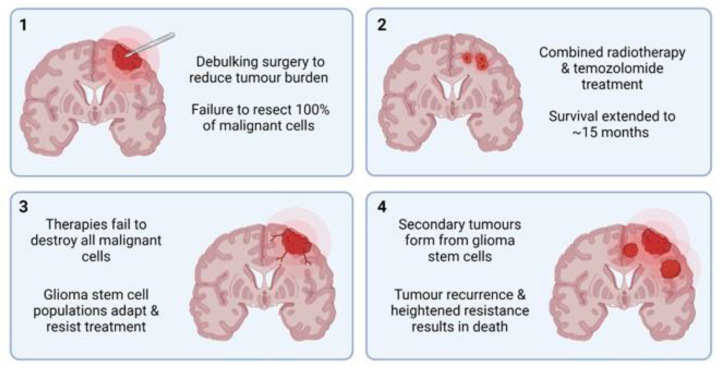
Overview of the current ‘gold standard’ GBM treatment strategy. **Panel 1:** Debulking surgery is performed to remove as much tumour mass as possible. Due to extensive infiltration, some metastatic cells inevitably remain. **Panel 2:** Combined radiotherapy and temozolomide (TMZ) treatment is administered over several months. **Panel 3:** Remaining malignant cells (primarily GSCs) overcome and adapt to TMZ treatment, continuing to proliferate. **Panel 4:** Inevitably, tumour recurrence and metastasis occur, resulting in eventual death.~: Median survival.

**Table 1 cancers-14-04767-t001:** Potassium ion channels in non-CNS cancers.

Channel	Cancer Type	Highlights	References
**Voltage-gated potassium channels**			
K_v_1.1	Breast	↑^1^ Expression relates to ↑ metastasis and tumourigenesis	[68]
	Prostate	↓ Expression in higher grade tumours, although variable between patients	[69,70]
K_v_1.3	Breast	↑ Expression regulates migration but not apoptosis or proliferation	[68]
	Breast	Inhibition reduced malignant cell proliferation	[71]
		↓ Expression in grade III tumours	[72]
		↑ Expression in patient samples and cell lines	[73]
	Colon	K_v_1.3 is a regulator of migration but not apoptosis or proliferation	[68]
	Leukemia	No observed relationship with malignancy, acts as a tumour suppressor	[74,75]
	Leiomyosarcoma	↑ Expression in more aggressive tumours	[76]
	Smooth muscle	↑ Expression in more severe phenotypes	[73]
K_v_1.5	Lymphoma	Expression reduces with increased malignancy	[77]
	Stomach	Involved with malignant cell proliferation via Ca^2+^ regulation	[78]
	Osteosarcoma	Inhibition halts proliferation via cell cycle arrest at G0/G1	[79]
	Cervical	Regulates cell cycle of malignant cells (works with K_v_9.3)	[80]
	Leiomyosarcoma	↑ Expression in more aggressive tumours	[76]
K_v_2.1	Stomach	Involved in malignant cell proliferation via Ca^2+^ regulation	[78]
	Lung	↑ Expression and regulates migration in more aggressive malignancies	[81]
K_v_3.4	Oral	Regulates invasion and tumourigenesis	[82]
	Breast	Inhibition results in ↓ cell proliferation	[83]
K_v_4.1	Colon	↑ Expression and role in cell proliferation	[84]
	Breast	↑ Expression in more severe phenotypes, knockdown inhibits proliferation	[83]
	Gastric	↑ Expression in human gastric cancer cell lines	[85]
K_v_7.1	Breast	Expression induces oncogenesis and growth	[67]
K_v_10.1	Stomach	Atypical expression and regulates proliferation	[86]
	Osteosarcoma	Inhibition results in ↓ cell proliferation via arrest at G1	[87,88]
**Calcium-activated potassium channels**			
K_Ca_1.1	Mesothelial	↑ Expression in more malignant phenotype, knockdown inhibits migration	[89]
	Sarcoma	Inhibition sensitised cells to paclitaxel, doxorubicin, and cisplatin	[90]
**K_Ca_2.3**	Colorectal	Forms a lipid raft ion channel complex with TRPC1/Orai1 to enhance migration, knockdown significantly reduced migration	[91]
	Breast	↑ Expression in highly metastasizing cell lines, knockdown greatly reduced migration	[92]
K_Ca_3.1	Breast	↑ Expression linked to lower overall survival	[93]
	Lung	Inhibition reduced tumour growth in vivo	[94]
	Renal	↑ Expression linked to lower overall survival and increased metastasis	[95]
	Endometrial	Inhibition reduced malignant cell growth in vitro and in vivo	[96]
	Pancreatic	Inhibition reduced malignant cell growth in vitro	[97]
**Inward-rectifying potassium channels**			
K_ir_2.2	Prostate, stomach, breast	Knockdown increased reactive oxygen species leading to cell cycle arrest	[98]
K_ir_5.1	Parathyroid	↑ Expression in parathyroid carcinoma	[99]
	Thyroid	↓ Expression in anaplastic thyroid carcinoma	[100]
	Pancreas	↓ Expression in pancreatic ductal adenocarcinoma (data set)	[101]
**Two-pore domain potassium channels**			
TREK-1	Prostate	↑ Expression in cancer vs. healthy prostate cancer tissue	[102]
TASK-3	Colorectal	↑ Expression in colorectal cancer samples	[103]
	Breast	Significant overexpression in 44% of breast tumours	[104]

^1^ ↑ Indicates high expression, ↓ indicates low expression.

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
