# Peer review of "Potassium Ion Channels in Malignant Central Nervous System Cancers"

_cancers, 2022, doi:10.3390/cancers14194767_

Round 1
Reviewer 1 Report
This paper reviews the relationship between potassium ion channel function and the progression of CNS cancers. The authors clearly illustrate the effects of different potassium ion channel subtypes on CNS cancers hallmark and provide some insights into possible therapeutic targets. However, there are still some problems, which must be solved before it is considered for publication. Therefore, my recommendation is a major revision.
1. In abstract, “ wherein treatment of the primary tumour leads to the development of treatment-resistant secondary tumours.” Confusing, please rewrite.
2. In abstract, the conclusions of this review do not come across as obvious.
3. For the first use of the abbreviated word, it need to be indicated in (parentheses), even though it appears in the abstract (Line 91, Line 225: CNS; Line 230: LGG)
4. What is the specific relationship between cellular electrical activity and DNA synthesis, please describe it. (Line 70)
5. Do “potassium ion channel” and “potassium channel” mean the same thing? If they are the same, please keep one statement throughout the paper.
6. “These channels are activated by increases in internal Na+ concentration”. Is the word "increases" right? (Line 145)
7. Replace the data of 2018 if the latest data is available. (Line 224)
8. Kv in Figure1 should be written in accordance with “VGKC” in the main text.
9. Section 3.1&3.2 only described HGG, but what about LGG and MB?
10. “The cell type of origin for GBM (Line 262~265)” is better to move to the general introduction of “Malignant CNS cancers”.
11. List some potassium channel antagonists approved for clinical use. (Line 527)
12. In Fig2, it is better to put the arrow symbol after the text. What’s more, the equal sign is not appropriate for a progressive relationship.
13. In Fig3, using numbers or letters to label images can make the figure easier to understand.
Reviewer 2 Report
The review is well documented and very clear. I have only few minor comments and suggestions.
Minor comments
P.5 line 183-184 the sentence is not clear regarding brain and myoblasts
P.7 line 205-211 and table 1 : the authors report involvement of KCa channels in potentiation of migration. They could report the work showing that SK3 channels are incomplex with calcium channels and promotes migration of breast cancer cells and colorectal cancer cells and is involved in metastasis (Chantome et al., Cancer Res. 2013; Gueguinou et al., Oncotarget 2016). in HCT-116 cells, SK3 enhances SOCE mediated by the Orai1/TRPC1 channel
complex. formation of lipid-raft Cav-1/Orai1/TRPC1 complex cooperating with SK3 to promote SOCE-dependent cancer cell migration (Gueguinou et al., Oncotarget 2016).
P.9 Regarding resistance to treatment the authors could mention that the pppulation of Cancer stem cells is involved in the process of resistance as well as in recurrence.
4.2 low grade glioma P13 line 451-458. Grade TT LGG present intratumoral cell heterogeneity and the authors report studies comparing astrocytoma and olygodendromas. An Interesting study in slow growing brain tumors, diffuse grade II IDH-mutant glioma, have reported that KCNN3 and electrophysiologically-active Ca2+-activated apamin-sensitive K+ channel (KCNN3/SK3) is a marker of astrocyte like tumoral cells (astrocytomas) in contrast to oligodendrogliomas (Augustus et al., Cancers, 2021).
Round 2
Reviewer 1 Report
This manuscript has been significantly improved after revision. I agree it is ready for publication.